# Pandemic response in pluralistic health systems: a cross-sectional study of COVID-19 knowledge and practices among informal and formal primary care providers in Bihar, India

Krishna D Rao ,[1] Japneet Kaur,[1] Michael A Peters,[1] Navneet Kumar,[2] Priya Nanda[3]

[1]Department of International Health, Johns Hopkins University, Baltimore, Maryland, USA
[2]Oxford Policy Management, Patna, India
[3]Bill and Melinda Gates Foundation India, New Delhi, Delhi, India

**Correspondence to**
Dr Krishna D Rao;
kdrao@jhu.edu

## ABSTRACT

**Objectives** Responding to pandemics is challenging in pluralistic health systems. This study assesses COVID-19 knowledge and case management of informal providers (IPs), trained practitioners of Ayurveda, Yoga and Naturopathy, Unani, Siddha and Homeopathy (AYUSH) and Bachelor of Medicine, Bachelor of Surgery (MBBS) medical doctors providing primary care services in rural Bihar, India.

**Design** This was a cross-sectional study of primary care providers conducted via telephone between 1 and 15 July 2020.

**Setting** Primary care providers from 224 villages in 34 districts across Bihar, India.

**Participants** 452 IPs, 57 AYUSH practitioners and 38 doctors (including 23 government doctors) were interviewed from a census of 1138 primary care providers used by community members that could be reached by telephone.

**Primary outcome measure(s)** Providers were interviewed using a structured questionnaire with choice-based answers to gather information on (1) change in patient care seeking, (2) source of COVID-19 information, (3) knowledge on COVID-19 spread, symptoms and methods for prevention and (4) clinical management of COVID-19.

**Results** During the early days of the COVID-19 pandemic, 72% of providers reported a decrease in patient visits. Most IPs and other private primary care providers reported receiving no COVID-19 related engagement with government or civil society agencies. For them, the principal source of COVID-19 information was television and newspapers. IPs had reasonably good knowledge of typical COVID-19 symptoms and prevention, and at levels similar to doctors. However, there was low stated compliance among IPs (16%) and qualified primary care providers (15% of MBBS doctors and 12% of AYUSH practitioners) with all WHO recommended management practices for suspect COVID-19 cases. Nearly half of IPs and other providers intended to treat COVID-19 suspects without referral.

**Conclusions** Poor management practices of COVID-19 suspects by rural primary care providers weakens

## Strengths and limitations of this study

► This is the first large-scale survey to document informal providers' and other private primary care providers' COVID-19 knowledge, and case management practices in India.

► The study was conducted during the COVID-19 pandemic in the state of Bihar, India, by contacting primary care providers by phone; as such, it provides insight into the practices of primary care providers during the pandemic.

► In low-income and middle-income countries (LMICs) like India which have a large presence of informal providers in the health workforce, rural COVID-19 suspects will likely first visit an informal provider; as such, study findings have important implications for pandemic control strategies in LMICs.

► This study is based on telephonic survey of primary care providers in Bihar and their stated practices may not fully reflect what they actually do in practice.

government pandemic control efforts. Government action of providing information to IPs, as well as engaging them in contact tracing or public health messaging can strengthen pandemic control efforts.

## INTRODUCTION

Many low-income and middle-income countries (LMICs) have pluralistic health systems where clinical care providers practice several systems of medicine.[1–3] This pluralism is further exacerbated by the presence of both formal and informally trained health workers, particularly in the primary care space.[3] In India, informal providers (IPs) are ubiquitous in the health workforce. IPs service a large share of outpatient visits and are typically the first contact providers for patients seeking ambulatory care in rural or poor urban

areas.[4 5] As such, patients in these areas with common coronavirus disease (COVID-19) symptoms such as fever, fatigue, cough or diarrhoea, would likely first visit an IP. During disease outbreaks there is concern about how well healthcare providers are informed about the disease and its management, particularly when knowledge about the disease is rapidly changing. During previous outbreaks, such as severe acute respiratory syndrome (SARS) and Middle East respiratory syndrome (MERS), collecting real-time information on provider knowledge and understanding of an emerging disease case management has supported response efforts by improving triage procedures and reducing infections in healthcare settings.[6–8] Like SARS and MERS, COVID-19 is a coronavirus disease, spread by airborne transmission from close personal contact. Despite the large presence of IPs in the primary care workforce of many LMIC countries, little is known about their knowledge and practices related to COVID-19.

IPs have no formal medical training from a recognised institution, though they commonly have some form of informal training.[3] In several countries, IPs are a substantial presence in the health workforce—they constitute around 88% of all healthcare providers in Bangladesh, and 77% in Uganda.[3] In India most ambulatory curative health services are provided by the private sector, and paid for out-of-pocket due to limited financial protection coverage, despite the fact that free care is available at government clinics.[9] India's pluralistic health system has a variety of formal and informal primary care providers—IPs, qualified AYUSH (AYUSH are trained practitioners of Indian systems of medicine and homeopathy, ie, Ayurveda, Yoga and Naturopathy, Unani, Siddha and Homeopathy) physicians and allopathic doctors. IPs comprise between 24% and 43% of the health workforce in India, and their share of the health workforce varies greatly between and within states.[10 11] IPs comprise 70% of the rural primary care health workforce, compared with 31% in urban areas.[11] Smaller scale state studies also indicate a large presence of IPs—for example, one study found that in a district in the state of Karnataka 74% of the clinical care providers were IPs, while in a district in Uttarakhand, their share was 79%.[4 11] In general, IPs are trusted community members who practice within villages and charge fees-for-services which are paid for out-of-pocket. People seek care from IPs for a number of reasons, including trust in the care IPs provide, proximity and lower cost relative to formally trained private providers.[12]

Indian IPs are a heterogeneous group of medical providers who can practice allopathic or Indian systems of medicines, or a mix of these.[4] IPs typically treat common illnesses like fever, diarrhoea and respiratory conditions and play an important role in referring cases to higher-level health facilities.[4] The few studies on IPs in India report a range of clinical experience—including certificate courses in allopathic and Indian systems of medicine, or apprenticeships with qualified doctors.[4] Studies on IP treatment practices find that they produce poor quality

care, though they might be knowledgeable about treatment protocols.[3] Interestingly, studies that have compared IPs with qualified primary care doctors find only small differences between them in protocol adherence, and no differences in the likelihood of giving a correct diagnosis or treatment.[13]

Over the past year, several studies have assessed healthcare provider knowledge and perceptions of COVID-19. These studies have focused on qualified providers and report good knowledge of COVID-19, though there are important gaps to be addressed as best practices evolve over time.[14] Studies among health providers in low-income and middle-income settings have echoed the broader literature by reporting good knowledge of COVID-19 symptoms and prevention, but gaps in knowledge of case management protocols.[15 16] In India, studies on the knowledge and practices of healthcare professionals such as doctors, medical residents, medical students and other formally trained health workers reported high levels of knowledge of COVID-19 symptoms, and preventive measures, but suggested lower levels of proficiency in terms of case management.[17–19] To the best of our knowledge, no study has attempted to document knowledge and practices related to COVID-19 among IPs. In general, previous studies during SARS and MERS outbreaks reported that healthcare providers had good understanding of disease symptoms and prevention but did poorly on following protocols for case management.[20–23]

In this study we aim to understand the knowledge and practices of formal and informal primary care providers—IPs, AYUSH physicians and allopathic doctors—related to COVID-19. Our study is based on a telephonic survey of primary care providers in the state of Bihar in eastern India. Understanding the knowledge and practices of IPs and other primary care providers in the context of the COVID-19 pandemic has important implications for the healthcare that communities receive, and more importantly, for the government's pandemic response.

## METHODS

This paper is based on a cross-sectional survey of primary healthcare providers in rural Bihar conducted via telephonic interviews. The survey was conducted from 1 to 15 July 2020, a period of rapid increase in the COVID-19 cases in the state.

### Setting

With a population of over 100 million and a Gross Domestic Product (GDP) per capita of US$640 (compared with the national GDP per capita of US$2099) Bihar is among India's resource poor states. Its residents are spread across 38 districts and some 45 000 villages, 88% of which are considered to be located in rural areas.[24] Although the state has made important gains in population health over the last few decades, it remains among the poorer performing states of India. Bihar's health system is under-resourced, including its human

resources for health workforce, which is operating at 1.5 health workers per 10 000 population, well under WHO's recommended 22.8 workers per 10 000 population.[11] Bihar's pluralistic health system is characterised by a large presence of IPs, particularly in rural areas. This shortage of health workers and dependence on informal providers has hampered Bihar's ability to deal with the COVID-19 outbreak.[25] At the time of this study, the first half of July 2020, Bihar was experiencing a rapid increase in the number of recorded COVID-19 cases. Confirmed cases increased from around 400 cases per day at the beginning of July to about 1300 cases per day by mid-July. Daily new cases continued to steadily increase to a peak of 3900 new cases in mid-August. From the beginning of July to the end of August, India experienced more than a threefold increase in cases, from approximately 19 000 to 70 000 new cases per day. This came even after India instituted one of the strictest national lockdowns in the world which lasted from mid-March until the end of May. Under the lockdown, people were restricted from leaving their homes and all transport services, educational institutions and hospitality services were suspended—violators were punishable by up to a year in jail. The lockdown severely affected the national economy and forced thousands of migrant workers to return to Bihar from cities across India. The spread of COVID-19 in rural Bihar has in part been attributed to the return of these migrant workers.[26]

## Study sample

The primary care providers in this study were identified from a parent household survey conducted in rural Bihar between November 2019 and March 2020. This parent survey's objective was to understand primary care seeking patterns in rural Bihar. In this survey, 70 blocks (of 534 total blocks) across Bihar's nine divisions were selected using stratified systematic random sampling. Within each block, five villages were selected using probability proportional to size sampling. In each village, a probability sample of 30 households was selected using segmented random sampling. The household survey covered 70 blocks and 343 villages across 37 districts in Bihar; a total of 8356 households, and 39 477 individuals were sampled. Of the individuals sampled, 15 811 (40%) reported being ill in the past month, and 10 617 (67%) of them sought care outside their home.

Respondents who sought care outside home were asked to report details of the providers they visited. We collected phone numbers and geolocations of these providers (if they were within 5 kilometres of the village) with the idea of surveying these providers at a later date to assess aspects of quality of care. Given the outbreak of COVID-19 in March 2020 in India, the provider survey had to be suspended. However, we felt that we could contribute to the state's COVID-19 response by contacting these providers via telephone to understand their experiences during the outbreak. Any provider identified through the parent study was eligible for inclusion in the telephone survey.

A total of 9497 provider contacts were recorded in the household survey. Of these, we had complete contact information for 6717 providers. After cleaning to remove drug shops (1603), community health workers (35) and duplicate providers (3941), we obtained telephone numbers of 1138 private providers across 256 villages. We made three attempts to contact each respondent. Of the total of 1138 providers contacted, we were able to successfully interview 522 private providers across 224 villages, achieving a 46% response rate. The key reasons for nonresponse included invalid telephone number (28%), respondent not interested in participating (19%), phone switched off (15%) and no response to calls (12%). At each block level primary health centre (PHC) in the 70 blocks covered by the household survey, the PHC medical officer was contacted and included in our study. Of the 70 PHCs contacted, we were able to conduct telephone interviews with 25 PHC medical officers, which translates to a 36% response rate.

## Data collection

Providers were interviewed using a structured questionnaire with choice-based answers to gather information on (1) change in patient care seeking, (2) source of COVID-19 information, (3) knowledge on COVID-19 spread, symptoms and methods for prevention and (4) clinical management of COVID-19. Where provider answers were ambiguous, enumerators were trained to probe the respondent to reach a clear answer, after which enumerators made a judgement on the most appropriate answer choice among the available selections. Given the challenge of keeping the respondent engaged in a telephonic survey, every attempt was made to keep the tool short and precise. Average time taken to complete an interview was around 20 min. The phone surveys were carried out by Oxford Policy Management, Delhi. All enumerators possessed a nursing degree, had prior experience in conducting quantitative interviews and were trained to conduct telephonic surveys using computer-assisted telephonic interviewing (CATI) software integrated with CSPro.[27] The CATI software displays the questionnaire on the screen of a tablet and the interviewer records the answers on the tablet during the interview. It also records the calls between the interviewer and provider enabling spot checks at a later date. Using CATI minimises information bias as the skip logic is already embedded in the questionnaire and data is automatically recorded in a data management platform, removing the need for double data entry.

## Data analysis

A random subset of the calls recorded were checked by a data manager to identify any errors and mismatches with the data entered in CSPro. Variables of interest were mostly categorical and for most questions, respondents could select more than one response option. Respondents were classified according to their self-reported medical training. A provider was classified as an IP if

they served as providers in a private facility and reported their training as any of the following—registered medical practitioner (RMP), no formal qualification, nurse, pharmacist, community health worker and a range of other non-degree qualifications. Providers who reported being trained in Indian systems of medicine were classified as AYUSH doctors. Providers who said they had a Bachelor of Medicine, Bachelor of Surgery (MBBS) or higher degree were classified as MBBS doctors. We conducted exploratory data analysis on the variables of interest to identify response patterns by provider types. Univariate and bivariate statistics were computed for variables of interest using two sample t-tests or $\chi^2$ tests of significance across provider types where necessary. Graphical analysis of the data was carried out. Missing data (responses from two IPs) was not included in the analysis. Each provider was treated as an independent observation. Statistical analysis was conducted using Stata V.14.[28]

### Patient and public involvement

Questions for the household survey were pre-tested on the general public, including patients, and were appropriately modified. Reports of where sick household members sought treatment enabled identification and recruitment of primary care providers for this study. Further, the questionnaire was tailored so that it would not place an excessive burden on providers.

### RESULTS

The 547 providers included in our sample were from 224 villages across 34 of Bihar's 38 districts. Our sample included 452 (83%) IPs, 57 (10%) AYUSH providers and 38 (7%) MBBS doctors. Of these, 522 (95%) were private and 25 (5%) were public providers (table 1). All public providers were located either at a primary healthcare facility or a community health centre.

Providers were asked if they were still seeing patients despite the nationwide lockdown. Among IPs, 73% reported seeing patients in person, 5% reported consulting patients over the phone and 22% reported not seeing patients in the last week. AYUSH doctors reported slightly higher rates of physical patient interactions, with 79% seeing patients, 4% consulting patients over the phone and 18% not seeing any patients in the last week. Among MBBS doctors, 86% were seeing patients

| Table 1 | Sample characteristics | | |
|---|---|---|---|
| | **Private providers** | **Public providers** | **Total providers** |
| MBBS | 15 | 23 | 38 |
| AYUSH | 55 | 2 | 57 |
| Informal provider | 452 | 0 | 452 |
| Total | 522 | 25 | 547 |

AYUSH, Ayurveda, Yoga and Naturopathy, Unani, Siddha and Homeopathy; MBBS, Bachelor of Medicine, Bachelor of Surgery.

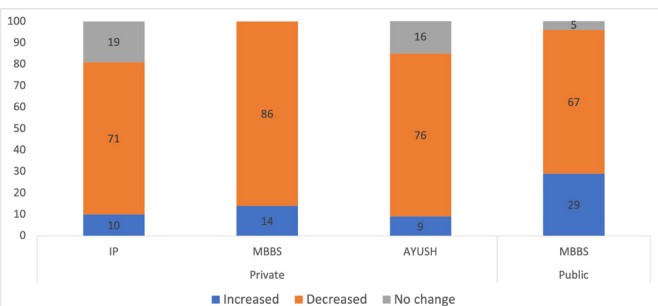

**Figure 1** Percentage of providers reporting change in patient visits during COVID-19 outbreak. AYUSH, Ayurveda, Yoga and Naturopathy, Unani, Siddha and Homeopathy; IP, informal provider; MBBS, Bachelor of Medicine, Bachelor of Surgery.

in person, 5% consulted patients over the phone and 8% did not see patients in the last week. A higher per cent of public MBBS doctors saw patients in the last week than private MBBS providers (91% vs 80%), and while a similar per cent of MBBS providers did not see any patients in the last week (9% of public vs 7% of private), 13% of private MBBS doctors consulted patients over the phone, while no publicly employed MBBS doctors employed this technique. We asked providers who were still seeing patients if there was any change in the volume of patient visits the week before the survey as compared with what they usually experience (figure 1). The majority of providers (72%), irrespective of qualification or public or private sector, reported a fall in patient visits. However, nearly one-fifth of IPs reported no change in patient volume over the prior week as compared with normal business. Public sector providers most frequently reported an increase in patient visits in the previous week.

A small (7%) proportion of private providers, including IPs, reported receiving training related to COVID-19 from either government or civil society sources, compared with 72% of public providers. Knowing where providers get their information on COVID-19 is important to plan future health communications activities (table 2). Television was the most common source of information for all provider types except public MBBS doctors, for whom it was the second most common information source. For IPs, AYUSH providers and private MBBS providers, newspapers were the second most common source of information. Nearly all (95%) of the public MBBS providers reported receiving information from government sources, compared with 29% of private MBBS providers, 34% of IPs and 35% of AYUSH doctors. Interestingly, mobile phones and the radio did not feature as important information sources across provider types.

We asked providers to name common COVID-19 symptoms (table 2). Overall, there was no significant difference in knowledge across provider types. The vast majority of providers in each group were able to identify symptoms such as fever, cough and breathing problems. In contrast, diarrhoea was far less frequently identified as a symptom of COVID-19 across provider types. Interestingly, nearly

**Table 2** Source of information and knowledge of COVID-19

| | IP (%) | Private MBBS (%) | Public MBBS (%) | AYUSH (%) | Total (%) |
|---|---|---|---|---|---|
| N (providers) | 452 | 15 | 23 | 57 | 547 |
| Sources of information | | | | | |
| Television | 76 | 80 | 52 | 74 | 75 |
| Newspaper | 53 | 53 | 35 | 40 | 51 |
| Government | 34 | 27 | 91 | 35 | 36 |
| Friends | 24 | 20 | 4 | 23 | 23 |
| Mobile phone | 12 | 7 | 4 | 7 | 11 |
| Radio | 9 | 7 | 9 | 14 | 10 |
| Knowledge of COVID-19 symptoms | | | | | |
| Fever | 90 | 100 | 96 | 89 | 90 |
| Cough | 83 | 93 | 87 | 86 | 84 |
| Breathing problem | 71 | 67 | 83 | 79 | 72 |
| Body ache | 24 | 13 | 35 | 25 | 24 |
| Sore throat | 23 | 7 | 17 | 26 | 23 |
| Fatigue | 15 | 13 | 9 | 12 | 15 |
| Diarrhoea | 6 | 13 | 13 | 9 | 7 |
| Loss or taste/smell | 8 | 13 | 30 | 5 | 9 |
| Knowledge of COVID-19 prevention | | | | | |
| Use face mask | 83 | 100 | 83 | 79 | 83 |
| Washing hands | 80 | 87 | 91 | 74 | 80 |
| Social distance | 76 | 67 | 87 | 79 | 77 |
| Stay at home | 15 | 33 | 17 | 18 | 16 |
| Avoid touching face | 11 | 7 | 17 | 12 | 11 |

Respondents can select multiple responses. There were two missing values (both IPs) for sources of information, or knowledge of COVID-19 symptoms, or knowledge of COVID-19 prevention.
AYUSH, Ayurveda, Yoga and Naturopathy, Unani, Siddha and Homeopathy; IP, informal provider; MBBS, Bachelor of Medicine, Bachelor of Surgery.

one-third of public MBBS providers identified loss of taste or smell as a COVID-19 symptom—a far greater percent than any other provider type.

Common public health measures for preventing COVID-19 infection, such as using a face mask, washing hands and distancing from other people were widely known across provider types (table 2). Others such as staying indoors or avoid touching one's face were less frequently reported.

Of interest is to know how primary care providers in Bihar would manage a suspected case of COVID-19. We asked providers 'In the past week, if a patient came to you with fever, cough, and breathing difficulty, what would you tell them to do?' According to the WHO ('Clinical Management of COVID-19: Interim guidance, May 27 2020') and Government of India (Clinical Management Protocol: COVID-19) guidelines, such a person would be a COVID-19 suspect having mild-to-moderate symptoms.[29 30] We classified provider responses in terms of the WHO-recommended actions for providers when presented with a suspected case of COVID-19 having mild-to-moderate symptoms (figure 2). The Government

of India guidelines had substantial overlap with these WHO-recommended actions. Overall, there was no significant difference between IPs and other provider types in following recommended actions, except for prescribing fever medication. A little more than half of the primary care providers said they would require the patient to wear a mask during consultation. The majority said they would tell the patient to take a COVID-19 test. Testing advice was highest for public MBBS (71%), followed by IPs (69%), AYUSH (63%) and private MBBS doctors (51%). Prescribing fever medication to the patient varied significantly across provider types, but most frequently prescribed by public MBBS doctors. Around 83% of public and 73% of private MBBS doctors said they would prescribe fever medication, compared with around half of the IPs and 36% of AYUSH providers. Advice about monitoring for complications was low across provider types—50% of private MBBS, 41% of AYUSH, 40% of IPs and 19% of public MBBS doctors said they would ask the patient to check if the symptoms became worse after a few days. Assessing patients for risk factors of severe complications, such as existing health conditions of heart disease

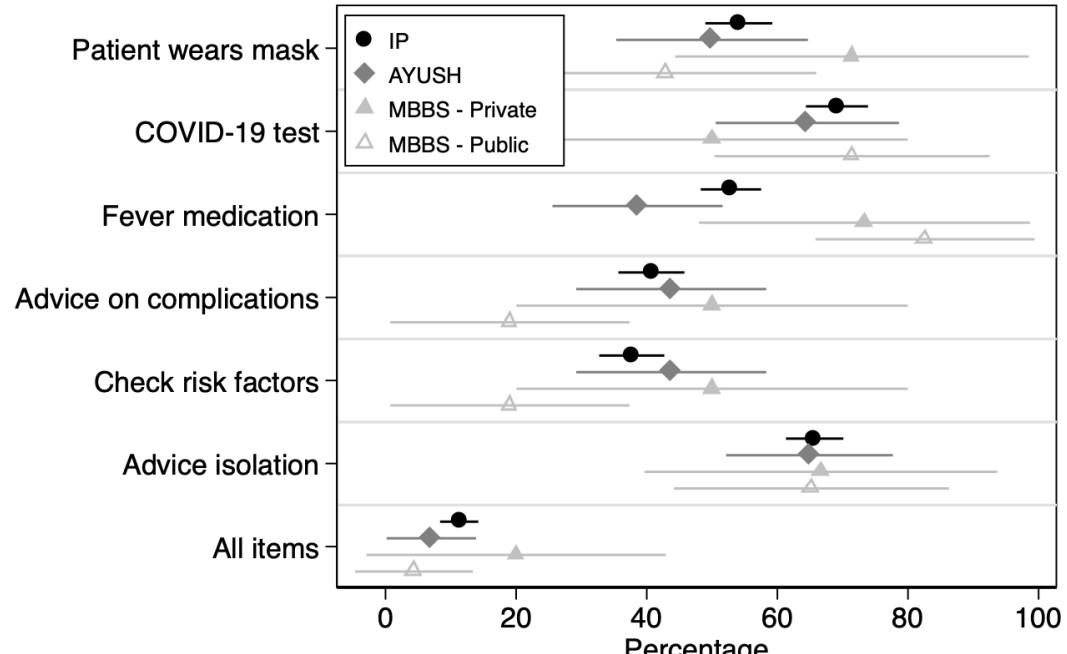

Note: (1) IP is Informal provider, MBBS is a provider with an MBBS degree, AYUSH is provider with a degree. in Indian systems of medicine or homeopathy.
(2) Figures represent point estiamtes and 95% confidence intervals.
(3) Recommendations from Clinical management of COVID-19, Interim guidance, WHO, May 27, 2020.

**Figure 2** Provider stated compliance with WHO recommended actions for COVID-19 suspects.

or diabetes, was reported by less than half of the providers in any group—50% for private MBBS, 41% AYUSH, 38% IPs and 19% public MBBS doctors. There was no significant difference across provider types in assessing patients for risk factors associated with complications. Advising patients to isolate at home was recommended by the majority of respondents with little difference between them. Overall, only 20% of private MBBS doctors, 11% of IPs, and 4% of 1pubic MBBS doctors and AYUSH practitioners reported all these recommended actions. On average, providers complied with 60% of the six recommended actions.

We asked providers if they would refer a patient who came to them in the past week with symptoms of fever, cough and breathing difficulty to a higher level health facility (figure 3). Nearly half of the providers in each group said they would not refer such patients. Across provider types, among those who said they would refer, government clinics or hospital were the preferred places for referral (66% of referrals were to government clinics or hospitals). There was no significant difference across groups in referral patterns.

## DISCUSSION

Responding to pandemics is particularly challenging in pluralistic health systems. When diverse systems of medicine are practiced and there is a mix of informal and formal health workers, it is a challenge to achieve uniform standards in providers' understanding of the

pandemic, ways to prevent infection and patient case management. This challenge is exacerbated by the ambiguous space that IPs occupy in India's health policy. One view, which reflects prevailing policy attitudes, is that IPs pose a danger to patients, and represent a problem that needs to be addressed. The alternative view is that they fill a vacuum in primary care service provision, and since they are already embedded within communities, it is pragmatic to engage with them. Findings from our study reflect both perspectives. Rural primary care providers as a whole were relatively well informed about the basics of COVID-19 symptoms and preventive measures, but performed poorly in terms of following recommended case management actions. In most cases, IPs performed similarly to MBBS or AYUSH doctors, but their low level of compliance could still endanger patients. On the other hand, over half of IPs recommended referring a suspect case to a government or other health clinic, so IPs could provide an important link to more sophisticated care. As the COVID-19 pandemic spreads across rural India, IPs will likely be the first contact providers for many patients; as such, there is much to be gained if appropriate actions are taken by them in patient encounters. Further, because IPs are embedded in rural communities, they can play an important role in contact tracing, and in public health messaging.

One of the significant, though not unexpected, findings from this study was the lack of contact (or training) that IPs and other private primary care providers had with

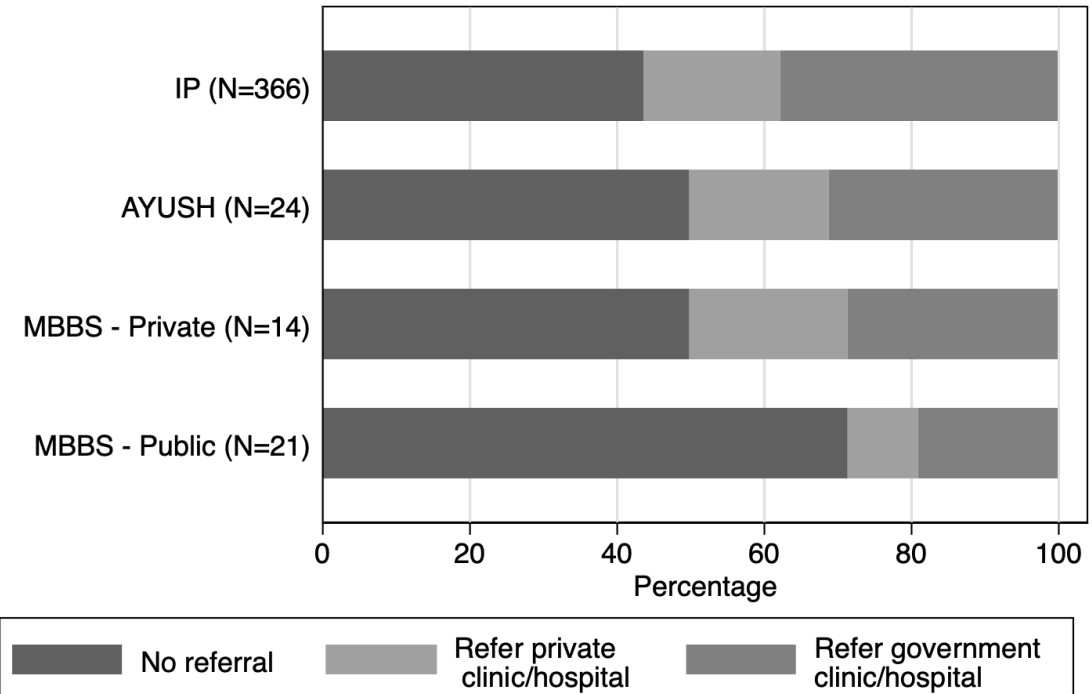

Note: Above resonses are to the question: In the past week if a patient came to you with symptoms of fever, cough, and breathing difficulty would you refer this patient to another doctor?
Question was not asked to providers who said they did not see patients in the last week

**Figure 3** Referral patterns for COVID-19 suspect cases. AYUSH, Ayurveda, Yoga and Naturopathy, Unani, Siddha and Homeopathy; IP, informal provider; MBBS, Bachelor of Medicine, Bachelor of Surgery.

government or civil society agencies. One consequence of this is that COVID-19 knowledge for most IPs and other primary care providers was primarily coming from television and newspaper sources. In contrast, most medical doctors in the government system reported receiving information on COVID-19 directly from government sources. Despite the lack of government engagement, IPs and other primary care providers were remarkably well informed of certain COVID-19 symptoms and preventive measures. This echoes findings from previous studies on the COVID-19 knowledge of qualified health professionals.[15 16] Importantly, this finding highlights the importance and responsibility of popular media sources in providing public health messaging to rural clinical providers. However, reliance on popular media alone may not be adequate. For example, providers had low awareness about symptoms, especially diarrhoea and the lack of taste or smell, which were described in the medical community as early as May 2020 and are now recognised as important COVID-19 symptoms.[31] Further, the importance of referral to clinics and testing sites could be further emphasised to better understand the local impact of the pandemic. While the frequency of referral between IPs and formal providers largely relies on established relationships and incentive structures, referral for COVID-19 testing could be an opportunity to strengthen linkages between the informal and formal sector.[4] As such, there is a role for government in providing health information to primary care providers, particularly in the context of a pandemic.

Engaging private practitioners embedded in local communities has been an important strategy for controlling the COVID-19 pandemic. In the urban slum of Dharavi in Mumbai, one of the largest slum areas of the world, local government agencies have effectively controlled the COVID-19 outbreak using a range of measures, including using local health practitioners to engage their communities with public health messaging, screening, contact tracing and providing clinical services.[32] Community trust in local healthcare providers considerably aided government efforts in the pandemic response. In Bihar, and elsewhere in India, there have been earlier efforts by government and civil society organisations to engage with IPs and other private sector providers to improve quality of care.[33] Such actions have not yet been taken for the COVID-19 response.[34] Clearly, providing information to IPs (and other private providers) on COVID-19, its prevention and where testing centres are located can help improve community knowledge on preventive practices and management of suspected cases. Since IPs are embedded in local communities, they are an invaluable resource for contact tracing and sharing public health messaging on COVID-19 to rural communities.

Our study finds that rural primary care providers had generally poor (stated) compliance with recommended case management practices for COVID-19. Their stated actions when presented with a suspected COVID-19 case, such as advising the patient to wear a mask, getting a COVID-19 test, prescribing fever medication, monitoring the patient for complications, asking patients about risk

factors and advising isolation were generally not practiced by IPs and more importantly, not by formal providers, as the onus is on the formally trained. In fact, only a small minority of primary care providers stated that they would do all these recommended practices. This points to the poor COVID-19 case management practices among formally trained providers and IPs alike. AYUSH and MBBS doctors, from both public and private facilities, did not perform significantly better than IPs. The observation that IPs (or AYUSH providers) do not differ significantly from MBBS doctors in their stated practices confirms similarities in quality of care that have been reported in other studies.[35] Findings from our study are likely generalisable to other rural contexts in resource limited states in India.

There are two notable limitations to our study. First, the high level of non-responses (55%), while common in telephone surveys, raises concerns about selection bias due to providers with certain characteristics not participating. There is some evidence for this—for example, among MBBS doctors there appears to be a higher non-response among public (75%) compared with private sector (46%) providers. If this non-response is related to the competency of respondents, that is, more competent doctors did not participate, then our estimates of compliance with quality actions would be biased in the negative direction. However, studies report that in physician surveys, the extent of bias due to non-response is likely minimal because physicians are quite homogeneous as a group in terms of knowledge and training, and variations that are present between them are unlikely to be associated with their likelihood of responding.[36] A second concern relates to the assessment of COVID-19 case management; which is based on what respondents said they would do. For one, reported actions could differ from what providers actually do in practice. Studies that have examined differences in knowledge and practice have found significant gaps, particularly among more knowledgeable providers.[37 38] While it is difficult to judge how well provider-stated intentions are reflected in practice, these stated actions can be viewed as an upper limit of what they might do in practice. Further, in the context of a telephone interview, it is possible that there was under-reporting of some key actions providers might take in practice because of trust issues or simply because of the nature of conducting interviews on the telephone. For example, it is somewhat surprising that so few IPs and AYUSH providers, who cannot officially prescribe allopathic medications, said they would prescribe fever medication (which is widely available) to someone with COVID-19 symptoms. While these providers could have answered that they prescribed other medicines such as cough medicines or antibiotics, about one-fifth of all IPs and AYUSH providers answered that they would not prescribe any medicines. Similarly, it is surprising that only a small proportion of public MBBS doctors said they would advise patients about complications, and risk factors.

In pluralistic health systems where IPs and other private providers comprise a large share of primary care

providers, embracing the entire health workforce in the government's COVID-19 response offers several advantages. Because IPs are likely to be the first contact primary care provider for the majority of symptomatic individuals, their management of suspect cases becomes vital to state and national efforts to control the outbreak. Moreover, because they are embedded within communities, IPs can assist in contact tracing, and public health messaging. As such, IPs can be an important partner in the government's COVID-19 response.

**Contributors** KDR, JK, MP and PN were responsible for conceiving and designing the study. JK and NK were responsible for organising data collection. KDR, JK and MP contributed to data analysis. All authors contributed to writing and editing the manuscript.

**Funding** Bill and Melinda Gates Foundation, Delhi. Grant # OPP119434. One of the authors (PN) is employed by the funder.

**Competing interests** PN is employee of the funding agency.

**Patient and public involvement** Patients and/or the public were involved in the design, or conduct, or reporting, or dissemination plans of this research. Refer to the Methods section for further details.

**Patient consent for publication** Not required.

**Ethics approval** Ethical approval was granted by the Sigma Institutional Review Board in India (Reference number: 1007/IRB/20–21) as well as by the Johns Hopkins University Institutional Review Board. The purpose of the study was explained and oral informed consent was obtained from each respondent. If the respondent agreed to participate, then a signed copy of the consent form was sent to the respondent via Short Message Service (SMS) text or WhatsApp.

**Provenance and peer review** Not commissioned; externally peer reviewed.

**Data availability statement** Data are available upon reasonable request. De-identified survey data is available from Krishna D Rao (kdrao@jhu.edu) on request.

**ORCID iD**
Krishna D Rao http://orcid.org/0000-0001-9347-3648

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
