## [Reviewer comments · BMJ Open]

ARTICLE DETAILS

TITLE (PROVISIONAL)	Pandemic Response in Pluralistic Health Systems: A cross-sectional study of COVID-19 knowledge and practices among informal and formal primary care providers in Bihar, India.
AUTHORS	Rao, Krishna; kaur, Japneet; Peters, Michael; Kumar, Navneet; Nanda, Priya

VERSION 1 – REVIEW

REVIEWER	Kaneko, Makoto Yokohama City University, Primary Care Research Unit, Graduate School of Health Data Science
REVIEW RETURNED	22-Dec-2020

GENERAL COMMENTS	Thank you for providing the opportunity to review the article. My comments as below are based on the STROBE checklist for observational study (http://www.equator-network.org/reporting-guidelines/strobe/) and I hope these comments improve your article. Major concerns #1. Please describe how to categorize the respondent's answer when the answer was ambiguous. Did the enumerators decide the category by themselves or discuss the issue with other enumerators? #2. Although the author stated "missing data were not included in the analysis", the proportion or characteristics of the missing data was not demonstrated. Please mentioned the information. #3. Did you obtain the information on the characteristics of the participants: age, sex or years of experience? If so, please describe them. Minor concerns #1 According to STROBE checklist #1, please add study design on the title or the abstract.
---

REVIEWER	Haruta, Junji University of Tsukuba
REVIEW RETURNED	30-Dec-2020

GENERAL COMMENTS	We believe that the findings revealed the rural primary care situation in India and the knowledge and management of COVID-
--

	19 among the study population, who included informal information providers (IPs). The findings may be transferable to rural primary care in low- and middle-income countries. However, as an original study, it has some issues and is difficult for readers to understand. Major comments 1. Title: Embracing pluralism for effective pandemic response The subtitle (COVID-19 knowledge and practices of informal and formal primary care providers in India) reflects the findings of the study. The authors' intent in Embracing pluralism was not clear from the study findings and discussion. Our understanding of the findings is that primary care providers are knowledgeable about COVID-19, but are not acting on it, and it is not different among IPs, AYUSH, and MBBBs. I recommend that the readers take a glance at the title and immediately understand the meaning of the title. 2. Introduction : 2-1. For readers to understand the healthcare system in India, it should be included that the penetration rate of the medical insurance system in India as a whole, the percentage of the population that has medical insurance, and the relationship between public and private medical institutions. Also, in the healthcare system of India, please describe the position and payments in consulting patients over the phone as described in the results. 2-2. Since the survey was conducted from July 1-15, it would be helpful to describe the progress of COVID-19 patients in Bihar from the end of June to July, so that readers can judge the consistency of the results. Also, if we could know the course of the patients in Bihar from the pandemic period to September, the readers could guess what the primary care providers estimated the pre-test probability of COVID-19. 2-3. Since lockdown is mentioned in the results, please describe in the intro how long the lockdown was in Bihar from mid-March. Also, since the severity of the lockdown control differs among countries, please mention the degree of severity in India. 3. Methods It would be better to describe how the structured questions were asked for the following four questions. Was the structured questionnaire an open question and/or a closed question and/or a choice-based question? (i) change in patient care seeking, (ii) source of COVID-19 information, (iii) knowledge on COVID-19 spread, symptoms, and methods for prevention, and (iv) clinical management of COVID-19. 4. Result 4-1. I think the following sentence corresponds to the Data analysis in Methods. P7 L21 "A provider was classified as IP if they served as clinicians in a private facility and reported their training as any of the following - Registered Medical A provider was classified as IP if they served as clinicians in a private facility and reported their training as any of the following - Registered Medical Practitioner (RMP), no formal qualification, diploma in modern and holistic medicine, nurse, pharmacist, community health worker, and a
--	--

	range of other non-degree qualifications. Providers who claimed to be trained in Indian systems of medicine were classified as AYUSH doctors, which is the official Providers who said they had a Bachelor of Medicine, Bachelor of Surgery (MBBS) or higher degree were classified as MBBS doctors, which is the official acronym for non-allopathic systems of medicine practiced in India. The results and discussion are mixed.” 4-2. There are a few statements that mix results and discussion. P8 L5 “MBBS providers official government communications were the most commonly cited knowledge sources after television. The difference seen in the MBBS group is due to the large number of public The difference seen in the MBBS group is due to the large number of public sector MBBS providers. The MBBS group comprises of 97% public and 3% private providers. Nearly 91% of the public MBBS providers reported receiving information from government sources, compared to 26% of private MBBS providers.” The above is an interpretation of the results. If you describe the results strictly, one possibility would be to show the results of the statistical analysis of public MBBS providers and private MBBS providers and add them as a Table. If the interpretation is based on descriptive statistics only, then it should be listed in Discussion. 4-3. I cannot understand which group the *Chi-sq test for group differences p-value < 0.05; listed in Table 1 was comparing. There seems to be a possibility of multiple comparisons. Please re-evaluate whether this statistical analysis is appropriate in a situation where multiple responses are possible. 5. Discussion In particular, I think major changes are needed in the Discussion. 5-1. From the results, it seems that there is not much difference in the knowledge and response to COVID-19 of AYUSH and MBBB as well as IP. Therefore, I think the discussion of IPs and others separately, especially in the first half, leads the reader in the wrong direction. For example, I think it would be more consistent with the findings of the study to discuss the response of rural primary care providers in India. If you can compare with other papers and discuss why the compliance of COVID-19 response in rural India is poor, it may be consideration that comes from the findings. Also, if you can provide comparable data on the percentage of referrals from IPs to clinics and hospitals in Bihar, it would help us compare COVID-19 with other diseases. Minor comments P6 L35 There is a typographical error in COVID-29 (→COVID-19). P5 L55 What does PPS sampling stand for? If it is an abbreviation for Probability Proportional to Size (PPS), it should be clearly stated in the first description. P6 L29
--	---

	Is there a difference between primary clinician and primary care providers? Some other words such as “doctors” means the same as “providers” confuse the readers. If they are used differently, please add an explanation so that the reader can understand it. P7 L8 As above, please indicate the abbreviation of SMS.
--	--

REVIEWER	Gautham, Meenakshi IDEAS Project, London School of Hygiene and Tropical Medicine, Global Health and Development
REVIEW RETURNED	08-Jan-2021

GENERAL COMMENTS	I like the study and the manuscript. It's an important study and a timely one. I would recommend accepting it with minor revisions if such an option were available for reviewers. I do have several comments, mainly to strengthen the Methods section and make the provider categorisations clearer, show distinctions between private and public MBBS in the findings, (as done in Table 1), and frame the comparisons between IPs and formal in the discussion in a way that first raises questions about the poor performance of formal MBBS and AYUSH. Currently it reads like poor case management is first a problem with the IPs and then with the others who have the qualifications. It should be the other way round. Since I am a bit short of time I have inserted all my comments in the pdf of the manuscript, rather than put them together in a word document. You should be able to open them with a single click. Otherwise can I request the editor to get them compiled into a single document, if necessary? Best wishes! - The reviewer provided a marked copy with additional comments. Please contact the publisher for full details.
--

VERSION 1 – AUTHOR RESPONSE

Reviewer	Comment	Response
# 1	Please describe how to categorize the respondent's answer when the answer was ambiguous. Did the enumerators decide the category by themselves or discuss the issue with other enumerators?	Thank you for raising this point. We have added the following sentence: “Where provider answers were ambiguous, enumerators were trained to probe the respondent to reach a clear answer, after which enumerators made a judgement on the most appropriate answer choice”
	Although the author stated “missing data were not included in the analysis”, the proportion or characteristics of the missing data was not demonstrated. Please mentioned the information.	Thank you for this comment- we have included an assessment of missing data in the comment of Table 2.
	Did you obtain the information on the characteristics of the participants: age,	Thank you for this query- we did not ask respondents in this survey for their age,

	sex or years of experience? If so, please describe them.	sex, or years of experience. We attempted to make the survey tool as short as possible and only kept the questions that we were sure to use in the write-up of this report. Thus, the focus was on the level of training of the provider (i.e. IP vs MBBS vs AYUSH) rather than any other demographic variable.
	According to STROBE checklist #1, please add study design on the title or the abstract.	Thank you for this suggestion- we have amended the title accordingly: "Pandemic Response in Pluralistic Health Systems: A cross-sectional study of COVID-19 knowledge and practices among informal and formal primary care providers in Bihar, India."
# 2	Title: Embracing pluralism for effective pandemic response The subtitle (COVID-19 knowledge and practices of informal and formal primary care providers in India) reflects the findings of the study. The authors' intent in Embracing pluralism was not clear from the study findings and discussion. Our understanding of the findings is that primary care providers are knowledgeable about COVID-19, but are not acting on it, and it is not different among IPs, AYUSH, and MBBBs. I recommend that the readers take a glance at the title and immediately understand the meaning of the title.	Thank you for raising this point and for your thorough review. We agree that the title can be further refined to match the findings of our study and are proposing the following changes: "Pandemic Response in Pluralistic Health Systems: A cross-sectional study of COVID-19 knowledge and practices among informal and formal primary care providers in Bihar, India."
	Introduction : For readers to understand the healthcare system in India, it should be included that the penetration rate of the medical insurance system in India as a whole, the percentage of the population that has medical insurance, and the relationship between public and private medical institutions. Also, in the healthcare system of India, please describe the position and payments in consulting patients over the phone as described in the results.	Thank you for this point, we agree that it is important to describe the extent of the medical insurance system and payment structure in the background section and have addressed this gap with the following additions: "In India most curative health services are provided by the private sector, though there is low health insurance coverage (about 13% of rural and 9% of urban population covered) and free care available at government clinics [9]." We did not ask about how payments were handled in phone consultation but have added additional information about how IPs are trusted community members who usually charge nominal fees in the background section:

		“In general, IPs are trusted community members who practice within villages and charge fees-for-services which are paid for out-of-pocket. People seek care from IPs for a number of reasons, including trust in the care IPs provide, proximity, and lower cost relative to formally trained private providers [12].”
	Introduction: Since the survey was conducted from July 1-15, it would be helpful to describe the progress of COVID-19 patients in Bihar from the end of June to July, so that readers can judge the consistency of the results. Also, if we could know the course of the patients in Bihar from the pandemic period to September, the readers could guess what the primary care providers estimated the pre-test probability of COVID-19.	Thank you for this suggestion, we have clarified information about the status of COVID-19 in Bihar and India more broadly in the new study setting section of the Methods: “At the time of this study, in the first half of July 2020, Bihar was experiencing a rapid increase in the number of COVID-19 cases. Confirmed cases increased from around 400 cases per day at the beginning of July to about 1,300 cases per day by mid-July. Daily new cases continued to steadily increase to a peak of 3,900 in mid-August. From the beginning of July to end of August, India as a whole experienced more than a three-fold increase in cases, from approximately 19,000 to 70,000 new cases per day.”
	Introduction: . Since lockdown is mentioned in the results, please describe in the intro how long the lockdown was in Bihar from mid-March. Also, since the severity of the lockdown control differs among countries, please mention the degree of severity in India.	Thank you for this suggestion, we also agree that more information was needed on the study setting and have included the following in the study setting section of the Methods: “This came even after India instituted one of the strictest national lockdowns in the world which lasted from mid-March until the end of May. Under the lockdown, people were restricted from leaving their homes and all transport services, educational institutions, and hospitality services were suspended- violators were punishable by up to a year in jail. The lockdown severely affected the national economy and forced thousands of migrant workers to return to Bihar from cities across India. The spread of COVID-19 in rural Bihar has in part been attributed to the return of these migrant workers [26].”
	Methods: It would be better to describe how the structured questions were asked for the following four questions. Was the	Thank you for this comment- our survey was a structured questionnaire with choice-based answer choices which were not read

	structured questionnaire an open question and/or a closed question and/or a choice-based question?	to the respondent by the enumerators. We have attempted to clarify this with the following additions to the methods section: “Providers were interviewed using a structured questionnaire with choice-based answers to gather information on...” “Where provider answers were ambiguous, enumerators were trained to probe the respondent to reach a clear answer, after which enumerators made a judgement on the most appropriate answer choice among the available selections” “Variables of interest were mostly categorical and for most questions, respondents could select more than one response option”
	Results: I think the following sentence corresponds to the Data analysis in Methods. P7 L21 “A provider was classified as IP if they served as clinicians in a private facility and reported their training as any of the following - Registered Medical A provider was classified as IP if they served as clinicians in a private facility and reported their training as any of the following - Registered Medical Practitioner (RMP), no formal qualification, diploma in modern and holistic medicine, nurse, pharmacist, community health worker, and a range of other non-degree qualifications. Providers who claimed to be trained in Indian systems of medicine were classified as AYUSH doctors, which is the official Providers who said they had a Bachelor of Medicine, Bachelor of Surgery (MBBS) or higher degree were classified as MBBS doctors, which is the official acronym for non-allopathic systems of medicine practiced in India. The results and discussion are mixed.”	Thank you for this suggestion- we agree with you and have placed this sentence into the sentence into the Methods section under Data analysis
	Results: There are a few statements that mix results and discussion. P8 L5 “MBBS providers official government communications were the most commonly cited knowledge sources after television. The difference seen in the MBBS group is due to the large	Thank you for this suggestion- we have reorganized the presentation of results to describe provider types in four categories: IPs, AYUSH doctors, MBBS private doctors, and MBBS public doctors. We have reformatted the table that presents this information (now Table 2), removed the

	number of public The difference seen in the MBBS group is due to the large number of public sector MBBS providers. The MBBS group comprises of 97% public and 3% private providers. Nearly 91% of the public MBBS providers reported receiving information from government sources, compared to 26% of private MBBS providers.” The above is an interpretation of the results. If you describe the results strictly, one possibility would be to show the results of the statistical analysis of public MBBS providers and private MBBS providers and add them as a Table. If the interpretation is based on descriptive statistics only, then it should be listed in Discussion.	mentioned interpretation of results (“The difference seen in the MBBS group is due to the large number of public sector MBBS providers.”), and have added the following interpretation: “For IPs, AYUSH providers, and private MBBS providers, newspapers were the second most common source of information. Nearly all (95%) of the public MBBS providers reported receiving information from government sources, compared to 29% of private MBBS providers, 34% of IPs, and 35% of AYUSH doctors.”
	Results: I cannot understand which group the *Chi-sq test for group differences p-value < 0.05; listed in Table 1 was comparing. There seems to be a possibility of multiple comparisons. Please re-evaluate whether this statistical analysis is appropriate in a situation where multiple responses are possible	Thank you for bringing this to our attention- we agree that this is confusing and likely not the most straightforward statistical analysis. We have removed from the Table in question
	Discussion: From the results, it seems that there is not much difference in the knowledge and response to COVID-19 of AYUSH and MBBB as well as IP. Therefore, I think the discussion of IPs and others separately, especially in the first half, leads the reader in the wrong direction. For example, I think it would be more consistent with the findings of the study to discuss the response of rural primary care providers in India. If you can compare with other papers and discuss why the compliance of COVID-19 response in rural India is poor, it may be consideration that comes from the findings.	Thank you for this suggestion. Aligned with your previous comment about making the title match the content of the article, we are keen on demonstrating the relative homogeneity in knowledge and practices between informal and formal primary care providers in a rural setting. Thus, we feel as though making the point about the role of IPs in pandemic response is crucial to our paper’s central theme. However, we do agree that we can be more clear about presenting results in terms of the overall response of rural primary care providers, rather than singling out IPs. We have made the following changes to reflect this stance: “Rural primary care providers as a whole were relatively well informed about the basics of COVID-19 symptoms and preventive measures, but performed poorly in terms of following recommended case management actions. In most cases, IPs performed similarly to MBBS or AYUSH doctors, but their low level of compliance

		could still endanger patients. On the other hand, over half of IPs recommended referring a suspect case to a government or other health clinic, so IPs could provide an important link to more sophisticated care.” “Our study finds that rural primary care providers providers had generally poor (stated) compliance with recommended case management practices for COVID-19”
	Discussion: Also, if you can provide comparable data on the percentage of referrals from IPs to clinics and hospitals in Bihar, it would help us compare COVID-19 with other diseases.	Thank you for this suggestion, information on this topic is generally sparse and varies greatly from community to community based on incentives and existing relationships. We have included the following: “Further, the importance of referral to clinics and testing sites could be further emphasized to better understand the local impact of the pandemic. While the frequency of referral between IPs and formal providers largely relies on established relationships and incentive structures, referral for COVID-19 testing could be an opportunity to strengthen linkages between the informal and formal sector [31]”
	Minor comments: P6 L35 There is a typographical error in COVID-29 (→COVID-19). P5 L55 What does PPS sampling stand for? If it is an abbreviation for Probability Proportional to Size (PPS), it should be clearly stated in the first description. P6 L29 Is there a difference between primary clinician and primary care providers? Some other words such as “doctors” means the same as “providers” confuse the readers. If they are used differently, please add an explanation so that the reader can understand it. P7 L8 As above, please indicate the abbreviation of SMS.	Thank you for your thorough review and for identifying these mistakes Amended, changed to COVID-19 Amended, changed from “PPS” to “probability proportional to size” There is no difference between primary clinician and primary care provider – we have removed instances of clinician to reduce confusion Amended, changed from “SMS” to “Short Message Service (SMS) text”

# 3	I do have several comments, mainly to strengthen the Methods section and make the provider categorisations clearer, show distinctions between private and public MBBS in the findings, (as done in Table 1), and frame the comparisons between IPs and formal in the discussion in a way that first raises questions about the poor performance of formal MBBS and AYUSH. Currently it reads like poor case management is first a problem with the IPs and then with the others who have the qualifications. It should be the other way round.	Thank you for your thorough review of our paper and for your invaluable comments- we have incorporated your suggestions and added a study setting subheading to the Methods section, clarified the categorizations of the providers in the data analysis section of the Methods, and also disaggregated findings throughout the results section. We have also considered your comment and fellow reviewer comments to restructure the discussion section to first address the main finding (that case management among rural providers is poor overall) and then delving into the role of IPs vs formally trained providers. Please see the following additions which reflect this change: “Rural primary care providers as a whole were relatively well informed about the basics of COVID-19 symptoms and preventive measures, but performed poorly in terms of following recommended case management actions. In most cases, IPs performed similarly to MBBS or AYUSH doctors, but their low level of compliance could still endanger patients. On the other hand, over half of IPs recommended referring a suspect case to a government or other health clinic, so IPs could provide an important link to more sophisticated care. As the COVID-19 pandemic spreads across rural India, IPs will likely be the first contact providers for many patients; as such, there is much to be gained if appropriate actions are taken by them in patient encounters. Further, because IPs are embedded in rural communities, they can play an important role in contact tracing, and in public health messaging.” “Further, the importance of referral to clinics and testing sites could be further emphasized to better understand the local impact of the pandemic. While the frequency of referral between IPs and formal providers largely relies on established relationships and incentive structures, referral for COVID-19 testing could be an opportunity to strengthen
------------	---	--

		linkages between the informal and formal sector [4].” “Our study finds that rural primary care providers had generally poor (stated) compliance with recommended case management practices for COVID-19. Their stated actions when presented with a suspected COVID-19 case, such as advising the patient to wear a mask, getting a COVID-19 test, prescribing fever medication, monitoring the patient for complications, asking patients about risk factors, and advising isolation were generally not practiced by IPs and more importantly, not by formal providers, as the onus is on the formally trained. In fact, only a small minority of primary care providers stated that they would do all these recommended practices. This points to the poor COVID-19 case management practices among formally trained providers and IPs alike. AYUSH and MBBS doctors, from both public and private facilities, did not perform significantly better than IPs. The observation that IPs (or AYUSH providers) don’t differ significantly from MBBS doctors in their stated practices confirms what has been reported in other studies [35]. Findings from our study are likely generalizable to other rural contexts in resource limited states in India.”
	Specific comments are in the attached PDF – One major comment to reorganize results in Table 1 to public/private Other comments are mainly on grammar, asking for further clarification, and improving references	Thank you for this suggestion- we have reorganized the results in the referred table to address public and private providers. Thank you for these detailed comments and for your time spent reviewing our paper- we have addressed all of the comments in the PDF document and improved references (ref 33, among others)

VERSION 2 – REVIEW

REVIEWER	Kaneko, Makoto Yokohama City University, Primary Care Research Unit, Graduate School of Health Data Science
REVIEW RETURNED	11-Feb-2021

GENERAL COMMENTS	The authors appropriately addressed my suggestions. Thank you for offering the opportunity to review the article for me.
--

REVIEWER	Haruta, Junji University of Tsukuba
-----------------	--

REVIEW RETURNED	28-Feb-2021
-------------

GENERAL COMMENTS	I have determined that this paper has been revised as per the reviewers' suggestions. The study seems to have provided a good picture of the current situation regarding COVID-19 in primary care in India. Thank you for your contributions.
---

REVIEWER	Gautham, Meenakshi IDEAS Project, London School of Hygiene and Tropical Medicine, Global Health and Development
-----------------	---

REVIEW RETURNED	27-Feb-2021
-------------

GENERAL COMMENTS	Thanks for addressing all the comments in your revised paper. Its reading very well and raises important issues. A couple of minor suggestions: Abstract: 1st para; 'trained practitioners of Indian systems of medicine (AYUSH)'. Suggest changing to 'trained practitioners of Ayurveda, Yoga and Naturopathy, Unani, Siddha and Homeopathy (AYUSH)'. Settings: 'With a population of over 100 million and a GDP per capita of US \$640, Bihar is among India's resource poor states'. (Can you also provide the all India GDP per capita to show how poor compared to the India average))
--

VERSION 2 – AUTHOR RESPONSE

Reviewer 3	Abstract: 1st para; 'trained practitioners of Indian systems of medicine (AYUSH)'. Suggest changing to 'trained practitioners of Ayurveda, Yoga and Naturopathy, Unani, Siddha and Homeopathy (AYUSH)'.	Revised abstract to meet journal formatting requirements. New sentence reads: “This study assesses COVID-19 knowledge and case management of informal providers (IPs), trained practitioners of Ayurveda, Yoga and Naturopathy, Unani, Siddha and Homeopathy (AYUSH), and allopathic medical doctors providing primary care services in rural Bihar, India.”
------------	---	--

Reviewer 3	Settings: 'With a population of over 100 million and a GDP per capita of US \$640, Bihar is among India's resource poor states'. (Can you also provide the all India GDP per capita to show how poor compared to the India average))	"With a population of over 100 million and a GDP per capita of US \$640 (compared with the national GDP per capita of US \$2,099) Bihar is among India's resource poor states."
------------	--	--